# Slow light in a 2D semiconductor plasmonic structure

Matthew Klein[1], Rolf Binder[1,2], Michael R. Koehler[3], David G. Mandrus[3,4,5], Takashi Taniguchi [6], Kenji Watanabe [7] & John R. Schaibley [1] ✉

Spectrally narrow optical resonances can be used to generate slow light, i.e., a large reduction in the group velocity. In a previous work, we developed hybrid 2D semiconductor plasmonic structures, which consist of propagating optical frequency surface-plasmon polaritons interacting with excitons in a semiconductor monolayer. Here, we use coupled exciton-surface plasmon polaritons (E-SPPs) in monolayer $WSe_2$ to demonstrate slow light with a 1300 fold decrease of the SPP group velocity. Specifically, we use a high resolution two-color laser technique where the nonlinear E-SPP response gives rise to ultra-narrow coherent population oscillation (CPO) resonances, resulting in a group velocity on order of $10^5$ m/s. Our work paves the way toward on-chip actively switched delay lines and optical buffers that utilize 2D semiconductors as active elements.

Coherent population oscillations (CPOs) have been used in atomic vapors and III–V quantum well semiconductor structures to generate spectrally narrow features in a medium's index of refraction that results in slow light[1–5]. CPOs originate from the interference of two driving fields (lasers) acting on an optical transition that gives rise to a modulation of the excited and ground state populations at the optical difference frequency between the pump and probe fields. The spectral width of the CPO resonance in a two-level system is given by the excited state lifetime. In a solid state system, such as a transition metal dichalcogenide (TMD) monolayer studied here, the spectral width of the resonance of the CPO is usually determined by the longest lifetime state that is coupled to the optical transition[6], which can be orders of magnitude longer lifetime (narrower line width) than the exciton dephasing time[4,6–8]. Monolayer TMD semiconductors host excitons which interact strongly with light and have been the focus of intense study over the past decade[9–14], leading to the rise of optoelectronic and plasmonic devices at the atomically thin limit[15–20]. It was previously shown in far-field optical measurements, that excitons in monolayer TMDs exhibit CPOs with narrow (few μeV) linewidths[6] offering the possibility to realize CPO-induced slow light via monolayer TMDs.

However, the atomically thin nature of monolayer TMDs limits their applications to optical propagation effects such as slow light, since in the typical optical configuration, the propagation vector is perpendicular to the two-dimensional (2D) layer, resulting in effectively zero (0.7 nm) interaction length. To overcome this limitation, we make use of a hybrid 2D material plasmonic structure consisting of a hexagonal boron nitride (hBN)-encapsulated $WSe_2$ monolayer transferred on top of a metallic waveguide that supports the propagation of surface plasmon polaritons[16] (SPPs) along the layer.

Recent measurements on 2D semiconductor plasmonic structures demonstrated coupling to the dark (out-of-plane) exciton[21–23], electrically tunable exciton–plasmon coupling[19,24], and nonlinear plasmonic modulation[16,25]. Previously, we developed a coupled exciton–SPP (E-SPP) model to explain the nonlinear response of the E-SPP in non-degenerate pump probe spectroscopy[16]. We emphasize that the E-SPP is a hybrid mode coupling 2D excitons to propagating SPPs over a relatively long (several micron) interaction length which is not possible in traditional far-field optical measurements, where a light beam is perpendicular to the 0.7 nm thick TMD monolayer. Specifically, we use a 2D semiconductor plasmonic structure to realize an

[1]Department of Physics, University of Arizona, Tucson, AZ 85721, USA. [2]Wyant College of Optical Sciences, University of Arizona, Tucson, AZ 85721, USA. [3]Department of Materials Science and Engineering, University of Tennessee, Knoxville, TN 37996, USA. [4]Materials Science and Technology Division, Oak Ridge National Laboratory, Oak Ridge, TN 37831, USA. [5]Department of Physics and Astronomy, University of Tennessee, Knoxville, TN 37996, USA. [6]International Center for Materials Nanoarchitectonics, National Institute for Materials Science, 1-1 Namiki, Tsukuba 305-0044, Japan. [7]Research Center for Functional Materials, National Institute for Materials Science, 1-1 Namiki, Tsukuba 305-0044, Japan. ✉e-mail: johnschaibley@arizona.edu

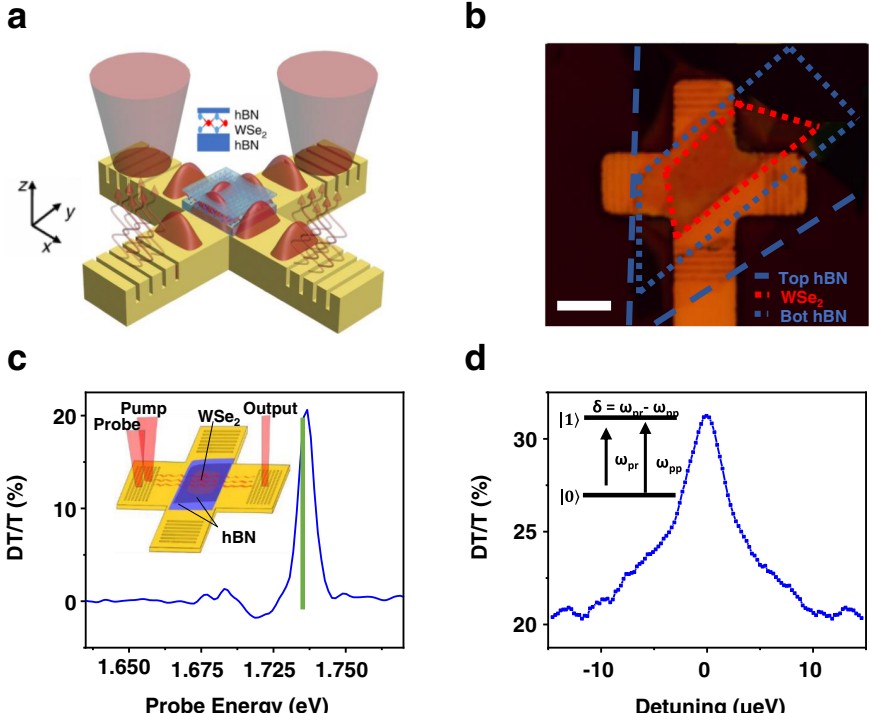

**Fig. 1 | Coherent population oscillations of excitons in 2D plasmonic structures. a** Depiction of the 2D material plasmonic structure. Surface plasmon polaritons (SPPs) are launched at the input of the device by focusing a free space laser onto either/both input couplers. The SPPs propagate through the waveguide where they interact with excitons in the WSe$_2$ monolayer, encapsulated in hBN. The SPPs are coupled back to free space photons by the output gratings. The upper inset depicts the hBN-WSe$_2$-hBN heterostructure on top of the waveguide. The axes used for the theory and simulations are shown. **b** Optical image of the main device used in the experiments. Scale bar 5 μm. The red dotted lines show the active region of the WSe$_2$ and blue dashed and dotted lines show the outlines of the top and bottom hBN, respectively. **c** Broad normalized differential transmission spectra (DT/T) for co-propagating SPP excitation. The green line denotes where the WSe$_2$ is pumped for **d**. The inset shows the schematic for co-propagating SPPs. **d** High-resolution DT/T spectra for co-propagating SPPs being pumped at 1.731 eV (716 nm). The inset depicts the pump and probe coupling to the $X^0$ neutral exciton state where $\omega_{pr}$ is the probe frequency, $\omega_{pp}$ is the pump frequency, and $\delta$ is the detuning between the pump and probe frequencies.

SPP–TMD interaction length of $L = 3$ μm and demonstrate slow light via nonlinear plasmonic population oscillations. Our structure design consists of an hBN encapsulated WSe$_2$ monolayer transferred on top of an optically thick gold waveguide in a "cross" shape with input and output grating couplers allowing efficient coupling (~10% per coupler) between free space photons and SPPs (see "Methods"). We use a "cross" waveguide structure in this work because it allows us to control the propagation directions of the pump and probe SPPs. Recall that SPPs have both an out-of-plane and in-plane polarization component, i.e.: $\hat{x}e^{i(k_{SPP}x-\omega t)} + \hat{z}e^{i(k_{SPP}x-\omega t)}$ for the $x$-propagating SPPs, and $\hat{y}e^{i(k_{SPP}y-\omega t)} + \hat{z}e^{i(k_{SPP}y-\omega t)}$ for the $y$-propagating SPPs where $k_{SPP}$ is the SPP wave vector and $\omega$ is the angular frequency. Here, we investigate the nonlinear response of the WSe$_2$ bright exciton whose optical dipole is in-plane and therefore couples to the "$x$" and "$y$" components of the SPPs. Therefore, by studying co-propagating pump and probe SPPs, we measure a signal analogous to the co-linearly polarized far-field response, and cross-propagating SPPs measures the cross-linearly polarized response.

## Results

### CPOs of E-SPPs

The 2D material plasmonic structures (depicted in Fig. 1a) were measured with linear and nonlinear two-color continuous wave (CW) pump-probe spectroscopy (see "Methods"). Figure 1b shows an optical microscope image of the structure with 2D material layers outlined. Both CW lasers were broadly tunable Ti:sapphire lasers with narrow line widths on order of 50 kHz. The pump and probe lasers were coupled into the waveguide generating SPPs that propagated through the TMD layer region. SPPs were then coupled back to free space

photons using the output gratings. A confocal pinhole was used before detection with a silicon photodiode to isolate light coupled out from a specific output grating. The linear transmission spectrum of the structure is shown in Supplementary Fig. 1. To characterize the slow-light structure, we maximized the nonlinear response by first measuring the differential transmission (DT) spectrum (Fig. 1c), i.e., the pump induced change to the probe transmission, with a pump laser photon energy of 1.725 eV. The raw DT signal was recorded using a lock-in amplifier and was then divided by the linear transmission signal (T) to yield the dimensionless DT/T response. The DT/T spectrum shown in Fig. 1c was recorded using 400 μW pump and probe powers (free space laser power incident on the coupler), in the co-propagating configuration as depicted in the Fig. 1c inset. We attribute the primary resonance near 1.73 eV with DT/T magnitude of ~20% to the neutral exciton ($X^0$)[26]. The weaker signals near 1.71 and 1.70 eV are identified as the charged exciton resonances[27,28], and the signal at 1.68 eV is the dark exciton[29]. In this work, we focus on the nonlinear response of the neutral exciton (near 1.73 eV) since it exhibits the largest nonlinear response, which is advantageous for large slow light effects via CPO[4].

Nearly degenerate two-color nonlinear spectroscopy was then performed by fixing the pump photon energy and scanning the probe photon energy through resonance with the pump at high resolution over a scan range of ~30 μeV (~8 GHz). Figure 1d shows the high-resolution DT/T response as a function of pump–probe detuning $\delta$ (probe energy—$\omega_{pr}$ minus pump energy—$\omega_{pp}$), recorded on the low energy side of the neutral exciton resonance (1.725 eV), for co-propagating SPPs. The DT/T response shows a narrow 6.8 μeV (full width half maximum) resonance with an additional amplitude of ~10% sitting "on top" of the broader DT/T resonance (shown in Fig. 1c)

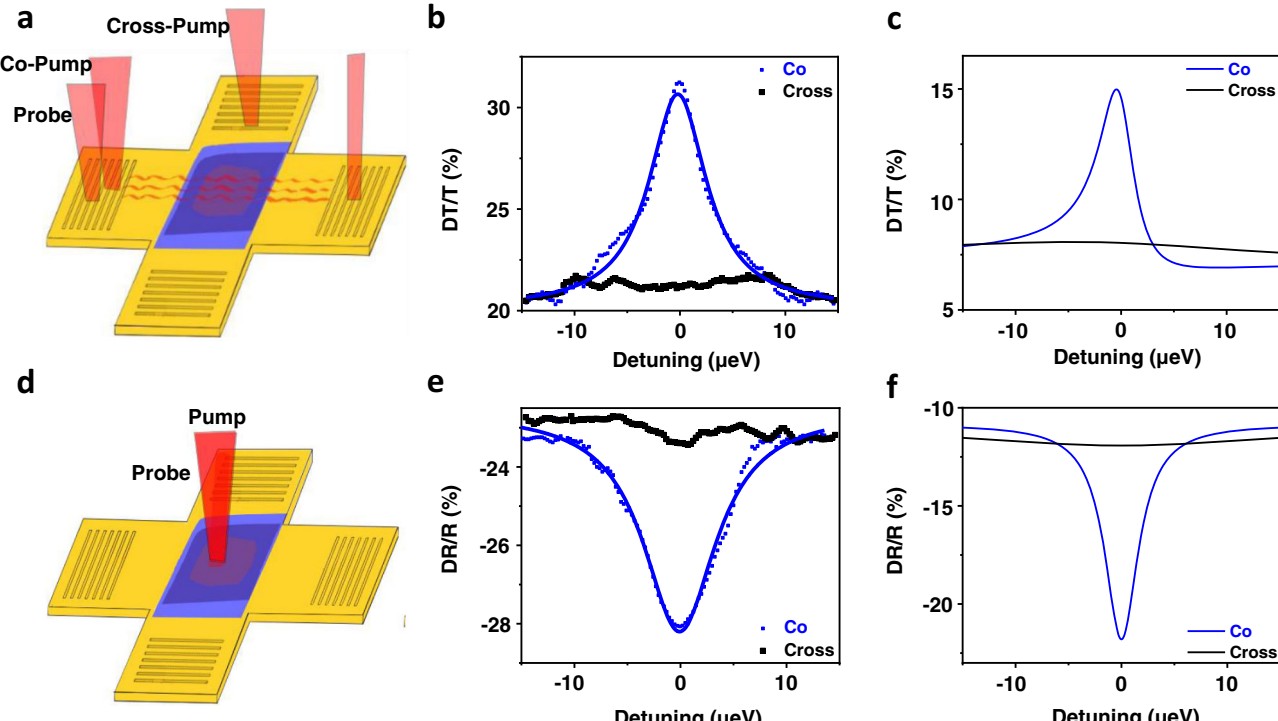

**Fig. 2 | Polarization dependence of the coherent population oscillation effect.** **a** Depiction of the surface plasmon polariton (SPP) pump and SPP probe configuration. **b** High-resolution normalized differential transmission spectra (DT/T) for SPP pump and SPP probe. The blue points correspond to co-propagating which are with to a Lorentzian (blue curve), and the black correspond to cross-propagating. **c** Theoretical DT/T for SPP pump and SPP probe. The blue curve corresponds to co-propagating, and the black corresponds to cross-propagating. **d** Depiction of the optical pump and optical probe configuration. **e** High-resolution normalized differential reflection spectra (DR/R) for optical pump-optical probe. The blue points correspond to co-polarized which are fit with a Lorentzian (blue line), and the black correspond to cross-polarized. **f** Theoretical DR/R for optical pump and optical probe. The blue curve corresponds to co-polarized and the black curve for cross-polarized.

around zero pump–probe detuning. This narrow spectral resonance is the signature of CPO of E-SPPs, similar to previous far-field measurements on excitons in monolayer MoSe$_2$[6].

The high-resolution measurements were repeated with different pump and probe configurations to explore the dependence of the CPOs on the pump polarization. Figure 2a, d show the experimental depictions for SPP pump and SPP probe as well as optical pump and optical probe respectively. Figure 2b (c) shows the experimental (theoretical) SPP pump, SPP probe case where the blue curves correspond to co-propagating pump and probe, and the black is cross-propagating pump and probe. Figure 2e (f) shows the experimental (theoretical) optical pump, optical probe case where the blue curves correspond to co-linearly polarized pump and probe, and the black is cross-linearly polarized pump and probe. In both cases, there is a narrow CPO resonance in the co-propagation/polarized (blue) cases and no significant resonance for the cross-propagation/polarized cases (black). The resonances in Fig. 2b, e were fitted to a weighted sum of the real and imaginary parts of a Lorentzian function with a linear offset yielding linewidths of 6.8 and 8.3 μeV for Fig. 2b, e, respectively. The similar linewidths and amplitudes (within a factor of 2) of the CPO resonances for both experiment and theory show good agreement. Similar data on a different device is shown in Supplementary Figs. 2a–f.

### E-SPP CPO theory

In order to explain the polarization dependence of the observed CPO resonances, we developed a theoretical model based on previous optical work on monolayer TMDs[6] and E-SPP[16]. To do this, we model the TMD excitons as Lorentz oscillators a small distance above a gold substrate to find the E-SPP dispersion relation in the linear and nonlinear regime. The excitonic nonlinearity, including its dependence on the pump and probe polarizations, is computed in third-order perturbation theory along the lines of ref. 30, here assumed to be dominated by phase-space filling. The detailed expressions for the nonlinear susceptibility are given in the Supplementary Note 1. Figure 2c(f) show the theoretical CPO response corresponding to probing with SPP (optical) and pumping with SPPs (optical). In this configuration, the co-polarized theory shows a CPO resonance, which disappears in the cross-polarized case (black), consistent with our experimental results. We note that the theoretical results of Fig. 2c could also be compared to an optical pump and SPP probe experiment since (within our approximation of neglecting out-of-plane excitons) only the squared magnitude of the in-plane electric field, present in both optical pump and SPP pump, contributes to the nonlinear susceptibility.

### Slow light from CPOs

We now turn to the slow light effect. The narrow CPO resonance results in a highly frequency dependent index of refraction and significant reduction in the E-SPP group velocity. To demonstrate slow light, we followed the technique of the Ku et al.[4], and we measured the group velocity using a heterodyne Mach–Zehnder interferometer (MZI) set-up depicted in Fig. 3a. Here, the 2D semiconductor plasmonic structure is in one arm of the interferometer and the probe reference with a piezotranslator (PZT) in the other allowing for the relative phase to be controlled. By mixing the probe beam with a probe reference, we are able to measure the frequency dependent phase shift. It is shown that the difference signal between photodiode 1 ($D_1$) and photodiode 2 ($D_2$) is proportional to the cosine of the phase delay of the probe path[2]

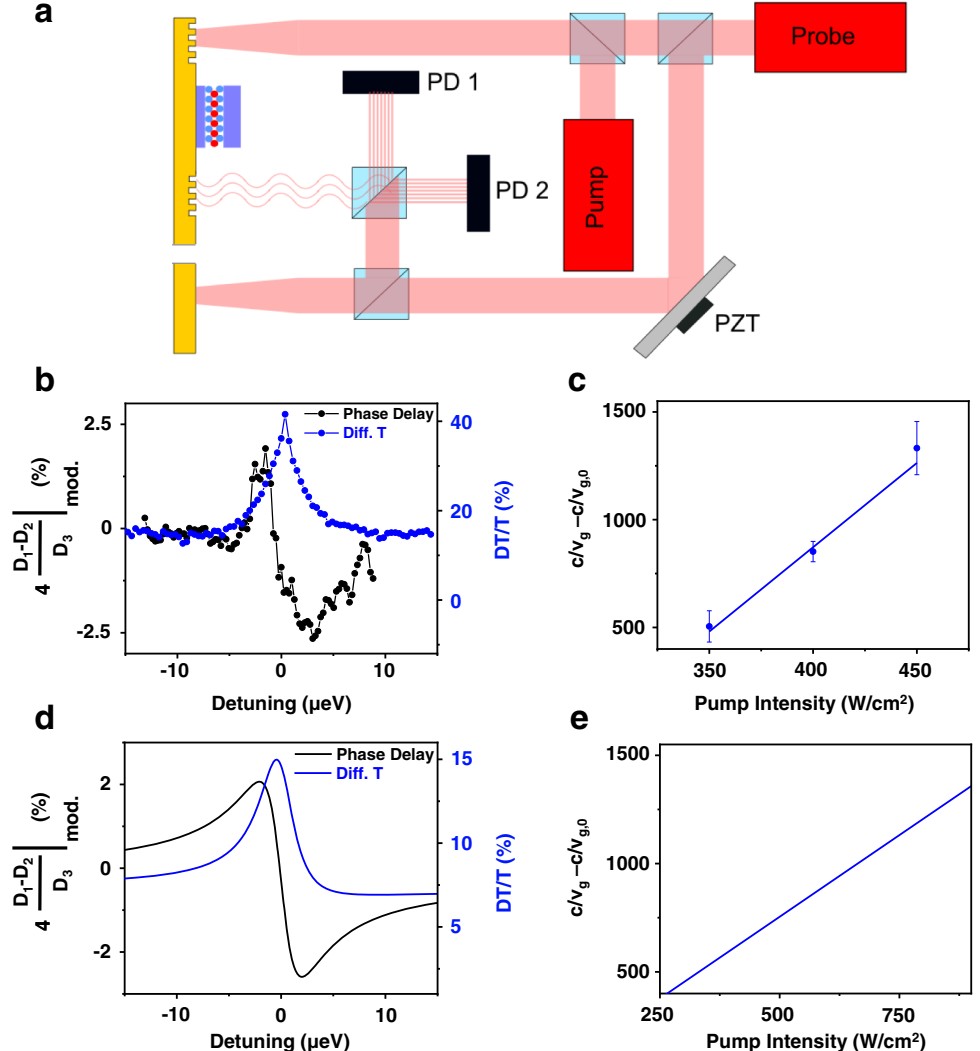

**Fig. 3 | Slow light via coherent population oscillations (CPOs). a** Depiction of the slow-light measurement. The probe is split off before the sample to create a reference beam path for one arm of the Mach–Zehnder interferometer (MZI). The reference beam path has a piezo translator (PZT) to control the phase of the heterodyne signal measured from photodiodes 1 and 2 ($D_1, D_2$). **b** Pump-induced phase delay MZI signal (black) from the CPO for 100 μW probe and 400 μW pump powers. The raw signal is scaled by a factor of 4 due to the modulation (subscript "mod") scheme used (see Supplementary Note 3). The normalized differential transmission (Diff. T) shown in blue shows the corresponding CPO DT/T resonance. **c** Group velocity slowdown ($c/v_g - c/v_{g,0}$) calculated from the phase delay for 100 μW probe power as a function of pump electric field strength at the WSe$_2$ layer ($E_p$), where $c-$speed of light, $v_g$–group velocity with the pump, and $v_{g,0}$–group velocity in absence of the pump. $c/v_{g,0}$ is approximately −0.4. The equivalent vacuum intensity is calculated via $I = \frac{1}{2}\epsilon_0 c|E_p|^2$ ($\epsilon_0$ = permittivity of free space) and is used as the horizontal axis in **c** and **e**. Error bars denote one standard deviation. **d** Theoretical plots of the normalized $D_1 - D_2$ signal (black) and DT/T (blue) that correspond to the data shown in **b**. **e** Theoretical intensity-dependent slowdown $c/v_g - c/v_{g,0}$.

(Supplementary Note 2):

$$D_1 - D_2 \propto e^{-\alpha L/2}\cos\left(\phi_{MZ} - \frac{\omega L}{c}n(\omega)\right) \quad (1)$$

where $n(\omega)$ is the index of refractive, $e^{-\alpha L/2}$ is the absorption of the TMD layer where $\alpha$ is the absorption coefficient and $L$ is its length (~3 μm), and $\phi_{MZ}$ is a phase difference in the interferometer, controlled by the PZT (see Supplementary Note 2). $\phi_{MZ}$ is set such that at zero detuning the $D_1 - D_2$ signal is zero. The phase delay as a function of $\phi_{MZ}$ is shown in Supplementary Fig. 3a, b. Figure 3b shows the measured population pulsation signal (blue) and the phase delay that can be inferred from the normalized MZI signal $D_1 - D_2$ for pump on minus pump off (black). We note that the signal is scaled by 4 to account for our modulation scheme (see Supplementary Note 3). These lineshapes also compare favorably to what our theory predicts for the population pulsation (blue) and the normalized $D_1 - D_2$ signal

(black) in Fig. 3d. For small detunings, we can use the small angle approximation to take the slope of the phase delay signal and estimate the group velocity of the material at different pump powers. We then take $c/v_g - c/v_{g,0}$ as a figure of merit for the slowdown of our system and plot it in Fig. 3c, where $c/v_{g,0}$ is the inverse group velocity without pump (in Supplementary Fig. 4, $c/v_{g,0}$ is approximately −0.4 at 0 detuning). For our highest available (vacuum equivalent) pump intensity (450 W/cm² in Fig. 3c), we measure a group velocity of $2.3 \times 10^5$ m/s, which corresponds to a slowdown factor of -1300. Using our theoretical approach, we obtain the E-SPP wave vector $k_x$ as a function of frequency directly and can thus calculate the group velocity as $dk_x/d\omega$ directly. We then show the slowdown as a function of the pump power in Fig. 3e. The detuning dependence of the group velocity in the absence of the pump for the E-SPP is shown in Supplementary Fig. 4 where the magnitude of the SPP or E-SPP group velocity never drops below 0.9$c$. This implies that the significant decrease in the group velocity of this system is entirely

due to the CPO effect, and only occurs in a narrow ~1 μeV bandwidth around the CPO resonance (Supplementary Fig. 5).

We find an (approximately) linear relationship between the slowdown and the pump power, both in theory within the $\chi^{(3)}$ approximation (third-order nonlinear response) and in the experiment. This suggests that the slowdown can be further increased, at least as long as the system is in the $\chi^{(3)}$ regime (the experimental value for the slowdown ($c/v_g$) per pump intensity, given by the slope in Fig. 3c, is 7.8 cm²/W).

We note that the asymmetry in Fig. 2c is due to the relative position of the exciton energy with respect to the SPP energy. In CPO measurements on other devices, the SPP probe signals were more asymmetric. There are even small changes to the symmetry of the signal based on the alignment of the beams into the gratings. It is very likely that the nuances of the device fabrication, material defects (our sample is slightly doped), and alignment cause the disagreement between the lineshapes of the experiment and the theory. We also expect that the quantitative agreement of the theory with the experiment could be further improved if in the theory deviations from Lorentzian lineshapes were taken into account, and other contributions to the nonlinear optical response, in addition to phase-space filling, were included.

## Discussion

In this work, we have demonstrated a significant slow light effect with 2D semiconductor excitons using an on-chip SPP waveguide. We used a nonlinear CPO resonance to demonstrate a slowdown of $c/v_g$ ~1300, limited by the available pump power in our experiment. We find that our slow down factor compares favorably to other reports including hBN phonons[31,32] ($c/v_g$ ~500), photonic crystals[33] ($c/v_g$ ~100), carbon nanotubes[34] ($c/v_g$ ~200), and stack of 15 GaAs/AlGaAs quantum wells[4] ($c/v_g$ ~30,000). We emphasize that the 2D semiconductor plasmonic structure used here exhibits a significant slow-down factor at optical frequencies and using only a single layer of a 2D semiconductor. We note that there are several possibilities available to increase the magnitude of the slow-down effect in 2D semiconductor plasmonic structures. As previously discussed[16], the nonlinear response of the E-SPP could also be enhanced by selecting materials whose exciton resonance is closer to the SPP resonance, or by using multi-layer TMD monolayer structures separated by hBN, which could increase the nonlinear response and resulting slow-down factor by orders of magnitude. There are also multiple solutions to improving the overall device quality by increasing the transmission of the waveguide such as by using directional grating couplers[35] or by using bow-tie couplers[36], which could increase the overall transmission to over 80%. Our work demonstrates the potential applications of TMD–plasmonic structures for optical buffers and other on-chip optical information processing applications that require control of the group velocity of light.

## Methods

### Optical measurements

The hybrid hBN-WSe₂-hBN/plasmonic structures were measured at 4.5 K in a closed-cycle optical cryostat (Montana Instruments) to reduce thermal broadening effects and enhance the nonlinear response. The transmission spectra, broad nonlinear, and high resolution measurements were performed using tunable CW Ti:sapphire lasers (MSquared SolsTiS). The lasers were focused to a spot onto the requisite input grating coupler. Light scattered from the probe output grating coupler was isolated using a spatial filter and detected with a silicon photodiode. In the linear transmission measurements, the probe laser was modulated for lock-in detection. In the nonlinear spectroscopy measurements, pump and probe beams were both modulated at different frequencies near 500 kHz to allow for lock-in detection at the modulation difference frequency. Due to our method of lock-in detection, we directly measure one fourth of the total

nonlinear response, which we correct for (see Supplementary Note 3). In the high resolution measurements, the pump and probe lasers were locked to external reference cavities, and the probe laser was fine scanned while maintaining reference cavity lock.

In the phase delay measurements, the probe signal was split into reference and probe beams where the reference beam path includes a mirror on a PZT to control the relative phase ($\phi_{MZ}$) between probe and reference. The probe beam was transmitted through the waveguide structure, and the reference beam was reflected off a separate gold pad before being interfered together. The interference signal was measured from both outputs of the MZI using a lock-in amplifier subtracting the $D_1$ and $D_2$ signals. In order to set $\phi_{MZ}$ such that the cosine function of Eq. 1 could be approximated by its argument, a voltage was applied to the PZT such that, at zero detuning between the pump and probe energies, the measured $D_1 - D_2$ signal was zero. The detuning spectra were then obtained as described above in the high-resolution measurements.

### Device fabrication

The gold waveguide was fabricated on 285 nm SiO₂/Si using a multistep lithography and etching process. The substrate is spun with S1813 photoresist and exposed using a maskless photolithography system and developed using MF-319. After the photolithography step, 200 nm gold was thermally evaporated onto the substrate using 10 nm titanium sticking layer. In the second lithography step, poly(methyl methacrylate) was spun and the grating pattern was written and developed using electron beam lithography. We used an Ar-based reactive-ion etching process to etch the grating couplers into the waveguide. The arms of the waveguide are 5 μm × 5.5 μm with a 5 μm × 5 μm central region. The grating couplers are composed of 5 grooves that are 60 nm deep with a width of 110 nm and period of 570 nm. The bare waveguide was characterized using atomic force microscopy and optical spectroscopy. The waveguide and grating coupler designs were optimized using a finite-difference time-domain (Lumerical) model. 2D materials were obtained via Scotch tape exfoliation from bulk crystals. The 2D heterostructure was fabricated and transferred onto the waveguide using a polymer-based technique[37].

## Data availability

The data that support the findings of this study are available in the Figshare database at the following link: https://figshare.com/projects/Slow_Light_in_a_2D_Semiconductor_Plasmonic_Structure/150330.

## Code availability

Codes used in this paper may be requested from the corresponding author.

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

## Acknowledgements

This work is primarily supported by AFOSR (Grant No. FA9550-20-1-0217). J.R.S. acknowledges additional support from the U.S. National Science Foundation (Grant Nos. ECCS-2054572 and DMR- 2054572), ARO (Grant No. W911NF2010215), and AFOSR (Grant No. FA9550-21-1-0219). R.B. acknowledges support from the U.S. National Science Foundation (Grant No. DMR-1839570). D.G.M. acknowledges support from the Gordon and Betty Moore Foundation's Epics Initiative, Grant GBMF9069. K.W. and T.T. acknowledge support from the Elemental Strategy Initiative conducted by the MEXT, Japan (Grant No. JPMXP0112101001) and JSPS KAKENHI (Grant Nos. 19H05790 and JP20H00354).

## Author contributions

J.R.S. conceived and supervised the project. M.K. fabricated the devices and performed the experiments. M.K. and J.R.S. analyzed the data with input from R.B. M.R.K. and D.G.M. provided and characterized the bulk $WSe_2$ crystals. T.T. and K.W. provided hBN crystals. R.B. developed and evaluated the theory. M.K., R.B., and J.R.S. wrote the paper. All authors discussed the results.

## Competing interests

The authors declare no competing interests.
