## [Peer Review File · Nature Communications]

Slow Light in a 2D Semiconductor Plasmonic StructureREVIEWER COMMENTS

Reviewer #1 (Remarks to the Author):

The paper builds up on an earlier PRL paper by some of the authors. In this paper (Ref. 6 from 2015), the authors have observed a spectrally narrow ($\sim \mu\text{eV}$) resonance in the nonlinear reflectivity spectrum of TMDC monolayers. The resonance was assigned to coherent population oscillations (CPO) of the TMDC excitons driven by the pump and probe pulses if they are detuned by less than the inverse of the exciton lifetime. Now the authors study the phase shifts imprinted upon the probe beam by such CPO. For this use grating coupling in a gold film to launch surface plasmon polaritons with different polarization properties. (The grating-coupler geometry has been used by the same group before, Ref. 16 from 2019). These SPPs propagate through a TMDC monolayer and the phase delay imprinted on the probe SPPs is detected in a Mach-Zehnder interferometer. The sharp resonance in the transmitted plasmon probe field due to CPO is resolved and a pump-induced change in probe transmission by up to 40% is demonstrated. The dispersive shape of the spectral phase of the transmitted probe SPP in the region around the CPO resonance is resolved and the group velocity of the SPP is deduced. An reduction in SPP group velocity by more than a factor of 1000 is reported. This is an impressive achievement.

I have several questions and comments:

- a) How large is the SPP group velocity in the absence of the pump. This is needed to interpret Fig. 3c – but I could not find a discussion of this.
- b) I presume that the value of the group velocity given in Fig. 3c is that at the frequency of the sharp resonance. I also presume that this dramatic reduction in group velocity appears only in the immediate vicinity of the sharp resonance. There is a figure showing a simulated color dependence of the group velocity in the supporting information, but the main does manuscript, apparently, does not address this crucial issue.
- c) The authors assign the sharp resonance to CPOs – but they never explicitly explain in the manuscript what these population oscillations are and where they come from. I think that much deeper discussion is needed in order to understand the experimental results.
- d) What is $\exp(-\alpha L/2)$ in the equation on page 6. This is not explained.
- e) The reduction in group velocity is observed in a $\sim \mu\text{eV}$ spectral region. I guess that limits the spatial of any modulator based on this effect to the nanosecond range. This is quite slow. Also, the overall transmission of the plasmon gratings is quite limited (few percent). What are the strategies for making the effects that are reported here interesting for practical applications?
- f) The authors state that they use pump intensities of a few 100 W/cm^2 . How many photons are needed to induce one switching event? Is there any realistic possibility to implement ultrafast plasmonic modulators without causing thermal damage to the sample?

In summary, the authors report a substantial reduction in SPP group velocity in a very narrow spectral region around the exciton resonance of a TMDC monolayer. The reported results seem interesting even though their practical relevance is most likely limited. I suggest that a final decision about the manuscript is made after the questions and comments that are raised above have been addressed by the authors.

Reviewer #2 (Remarks to the Author):

Overall this is an interesting paper that extends the current understanding of coherent population oscillations within TMDs for controlling excitonic effects and light confinement. The authors build on prior work where they illustrated SPP-excitonic modes in TMD monolayers. Here the authors extend this work to show that the hybridization can be used to control the coupled mode dispersion illustrating slow light and increased propagation lengths. Overall I find the paper interesting, but there are some confusing parts that need to be addressed. Further, it is still a little murky what

exactly is new other than using a different TMD and illustrating the ability to slow the SPP, which isn't surprising. They do point to an improvement in the propagation length (0.7nm to 3 μ m) with this approach, which is significant, but don't really discuss that after their intro, instead focusing on the slow down factor. I think in principle the paper could be improved for publication in Nature Comms if the below (and above) issues are addressed.

1. One of the principle findings in this work is this 1300-fold slowing of the SPP. This isn't necessarily that surprising considering that strong coupling between a localized and propagating mode will result in a hybrid dispersion with a propagation length and group velocity in between. This is well known and well reported. The fact that the slowing is so large is interesting to some degree, but considering the plasmon (very slow dispersion) is extremely fast and the exciton is effectively localized, this is what would be expected. The authors thus need to make a comparison to other systems to illustrate how large is this in contrast to the broader field of polaritonic strong coupling (e.g. vibrational polaritons I would assume would have even larger reductions) and then clearly state why this is beneficial other than generally stating these systems are good for on-chip delay lines and buffers.

2. In the caption for Figure 1 has two very different descriptions for panel d and calls to the cross polarized spectra that doesn't seem to be there.

3. I think the addition of propagating character is the more interesting finding. This of course is also not surprising to see, but much like SPP-Epsilon-near-zero coupling in prior works, this offers propagating character to a mode that is otherwise static, while still retaining its pertinent character. I would advise that this should be amplified in this discussion.

4. In Fig. 2c there is an asymmetry in the simulated detuning spectrum of the co-polarized configuration that does not show up in the experiment. Is there some insight into the origin of this? It doesn't seem to be simply an artifact of simulations since it is not present in Fig. 2f.

5. Finally, I understand that the authors are using the slow down factor to describe the reduction in the V_g in the hybrid SPP-E mode with respect to the uncoupled SPP, while the increase in the propagation length of the hybrid mode wrt the bare exciton. However, I don't believe as written this is going to be clear to the reader. I would advocate for the authors to either present a figure with the propagation lengths and group velocities for all three (bare SPP, bare exciton, hybrid mode) within the frequency range of interest and/or describe in the text how the hybridization provides some averaging of the two bare modes. As written it comes across as the group velocity is reduced and the propagation length increases, which are two contradictory statements without the stated context.

Reviewer #3 (Remarks to the Author):

This paper reports the slow light effect in a 2D semiconductor-plasmonic hybrid system. It's an interesting work. It claims the first demonstration of slow light in 2D semiconductors. The paper can be published given the authors address the following properly.

1. Why are the excitons in 2D semiconductors special for the slow light purpose? Does it require large exciton binding energy, or sharp resonances with long lifetime? From my understanding, it seems that only a sharp resonance is necessary to have the slow light and there are many possible candidates for such resonances, not limited to excitons.

2. How about IR active phonons for the slow light application? For instance, BN phonons.

3. Can the authors comment on the excitons in carbon nanotubes (1d system) for the slow light purpose? Can they achieve similar result as the authors demonstrated with 2D materials?

4. For TMDs, monolayer is a direct band gap material, but two or multilayer become indirect. However, they still have excitons at K-point. Are those excitons suitable for the experiment in the paper?

Response to Reviewer Comments for “Slow Light in a 2D Semiconductor Plasmonic Structure”

Reviewer #1 (Remarks to the Author):

Comment 1: The paper builds up on an earlier PRL paper by some of the authors. In this paper (Ref. 6 from 2015), the authors have observed a spectrally narrow ($\sim \mu\text{eV}$) resonance in the nonlinear reflectivity spectrum of TMDC monolayers. The resonance was assigned to coherent population oscillations (CPO) of the TMDC excitons driven by the pump and probe pulses if they are detuned by less than the inverse of the exciton lifetime.

Now the authors study the phase shifts imprinted upon the probe beam by such CPO. For this use grating coupling in a gold film to launch surface plasmon polaritons with different polarization properties. (The grating-coupler geometry has been used by the same group before, Ref. 16 from 2019). These SPPs propagate through a TMDC monolayer and the phase delay imprinted on the probe SPPs is detected in a Mach-Zehnder interferometer. The sharp resonance in the transmitted plasmon probe field due to CPO is resolved and a pump-induced change in probe transmission by up to 40% is demonstrated. The dispersive shape of the spectral phase of the transmitted probe SPP in the region around the CPO resonance is resolved and the group velocity of the SPP is deduced. An reduction in SPP group velocity by more than a factor of 1000 is reported. This is an impressive achievement.

I have several questions and comments:

Response 1: We thank the reviewer for their review of this manuscript and appreciate their supportive comments on our results.

Comment 2: How large is the SPP group velocity in the absence of the pump. This is needed to interpret Fig. 3c – but I could not find a discussion of this.

Response 2: We appreciate the reviewer’s advice on how to provide clarity to our paper. In the original manuscript, we failed to give a more in-depth discussion and provided only the inverse E-SPP group velocity in the Figure 3 caption. In the Supplement of the revised manuscript, we add the following two figures (Supplementary Figures 4 a,b) that clarify the issue raised by the reviewer. From the figures, we see that, without pump and without the exciton, the SPP group velocity in the vicinity of the exciton (but again, without the effect of the exciton) is $\sim + 0.9c$.

Including the excitonic effects, at the center of the exciton line (0 meV in Supplementary figure 4), the group velocity is $\sim - 2.4c$, corresponding to the value of the inverse group velocity (in units of c) of $\sim - 0.4$. On the μeV scale, the group velocity in the absence of the pump is constant. We note that the (negative) group velocity without the pump but with the exciton shows the usual behavior in the vicinity of a resonance, see for example the paper by Steven Chu¹. This behavior follows directly from the dispersion curve (which we have also included now in the Supplement) since the group velocity is just the derivative of the wave vector. We note that the value of $\sim - 2.4c$ is much larger than the value that we obtain with the pump, which is approximately $\sim + \frac{1}{1300} c \sim + 0.00076c$. There is also further discussion to this in our response to Reviewer 2’s Comment 5. We have added discussion of this to the first paragraph of page 7.

Supplementary Fig. 4. Theoretical E-SPP in absence of a pump. a, Dispersion relation of the E-SPP in absence of a pump. **b**, Group velocity of the E-SPP as a function of the detuning from the exciton resonance in absence of a pump.

Comment 3: I presume that the value of the group velocity given in Fig. 3c is that at the frequency of the sharp resonance. I also presume that this dramatic reduction in group velocity appears only in the immediate vicinity of the sharp resonance. There is a figure showing a simulated color dependence of the group velocity in the supporting information, but the main does manuscript, apparently, does not address this crucial issue.

Response 3: We thank the reviewer for bringing up the importance of the photon energy/color dependence of the slowdown. The group velocity is calculated at the center of the resonance where the reduction is at its maximum. The dramatic reduction in group velocity only occurs near the center of the CPO resonance with a bandwidth of $\sim 1 \mu\text{eV}$. We revised the paragraph of pages 7 to clarify this point.

Comment 4: The authors assign the sharp resonance to CPOs – but they never explicitly explain in the manuscript what these population oscillations are and where they come from. I think that much deeper discussion is needed in order to understand the experimental results.

Response 4: CPOs originate from the interference of two driving fields (lasers) acting on an optical transition which gives rise to a modulation of the excited and ground state populations at the optical difference frequency between the pump and probe fields. The spectral width of the CPO resonance in a two level system is given by the excited state lifetime. In a solid state system, such as a TMD monolayer studied here, the spectral width of the resonance of the CPO is usually determined by the longest lifetime state that is coupled to the optical transition², which can be orders of magnitude longer lifetime (narrower line width) than the exciton dephasing time. We have included a new discussion of CPOs to the first paragraph of main text.

Comment 5: What is $\exp(-\alpha L/2)$ in the equation on page 6. This is not explained.

Response 5: We thank the reviewer for noticing our omission. The factor $\exp(-\alpha L/2)$ is the absorption of the SPP due to the TMD layer where α is the absorption coefficient and L is the length of the TMD layer in the direction of the SPP propagation. We have since added its definition to page 7 of the main text.

Comment 6. The reduction in group velocity is observed in a $\sim\mu\text{eV}$ spectral region. I guess that limits the spatial of any modulator based on this effect to the nanosecond range. This is quite slow. Also, the overall transmission of the plasmon gratings is quite limited (few percent). What are the strategies for making the effects that are reported here interesting for practical applications?

Response 6: The Reviewer is correct that the $\sim\mu\text{eV}$ spectral width of the CPO resonance does limit the application of this slow down effect to the nanosecond range. Indeed, this is quite typical of other reported slow light effects in atomic systems and quantum wells since a spectrally narrow resonance is advantageous for realizing a large slow down effect, since the group velocity goes as $dn/d\omega$.

We emphasize that the design of our waveguide couplers is simple, and it works well for our purposes of studying E-SPPs; however, as the reviewer points out, in terms of practicality it is lacking. There are multiple solutions to improving the overall device quality such as using directional plasmonic couplers which could increase our coupling efficiency to 80%³. There has also been work that using bow-tie couplers to couple in and out of an Ag plasmonic waveguide that showed overall transmission exceeding 80%⁴. Either of these or other methods could be used to significantly increase the practicality of our device for future applications. We note that this overall efficiency transmission efficiency is not the goal of our present work and does not affect the reported slow light effect in any way. We have added a summary of these future approaches to page 9 the main text.

Comment 7: The authors state that they use pump intensities of a few 100 W/cm^2 . How many photons are needed to induce one switching event? Is there any realistic possibility to implement ultrafast plasmonic modulators without causing thermal damage to the sample?

Response 7: Since we have not performed a detailed quantum-optical analysis of the CPO effect (we are using a semi-classical theory in which only the semiconductor is treated quantum mechanically, while the light field is treated classically), it is difficult to predict exactly how many pump and probe photons are required to induce a single switching event. We will instead make an argument on our experimental results. Our current measurements use $\sim 10^{15}$ photons per second incident on the input grating for both the pump and probe excitations which leads to a dT/T signal on the order of 60%. Since DT/T scales linearly with the pump photons, the

number of pump photons can be reduced until a minimum acceptable threshold of DT/T is achieved. We appreciate the reviewers interest in the fundamental limits in our device's practicality as a nonlinear modulator; however, while interesting, this point is unfortunately beyond the scope of the present manuscript.

As to the reviewer's second point, for the powers we use in this manuscript there is no thermal damage to the sample. In a separate experiment not related to this manuscript we have been able to use pump intensities on the order kW/cm² without there being any thermal damage. So it is our predication that there should be no issues implementing our slow light effect due to limitations from thermal damage.

Comment 8: In summary, the authors report a substantial reduction in SPP group velocity in a very narrow spectral region around the exciton resonance of a TMDC monolayer. The reported results seem interesting even though their practical relevance is most likely limited. I suggest that a final decision about the manuscript is made after the questions and comments that are raised above have been addressed by the authors.

Response 8: We appreciate the author's support of the value of our work and hope that the revised manuscript is suitable for publication.

Reviewer #2 (Remarks to the Author):

Comment 1: Overall this is an interesting paper that extends the current understanding of coherent population oscillations within TMDs for controlling excitonic effects and light confinement. The authors build on prior work where they illustrated SPP-excitonic modes in TMD monolayers. Here the authors extend this work to show that the hybridization can be used to control the coupled mode dispersion illustrating slow light and increased propagation lengths. Overall I find the paper interesting, but there are some confusing parts that need to be addressed. Further, it is still a little murky what exactly is new other than using a different TMD and illustrating the ability to slow the SPP, which isn't surprising. They do point to an improvement in the propagation length (0.7nm to 3 um) with this approach, which is significant, but don't really discuss that after their intro, instead focusing on the slow down factor. I think in principle the paper could be improved for publication in Nature Comms if the below (and above) issues are addressed.

Response 1: We thank the author for their comments on our work. We note that the novelty of our work is the demonstration of a large 1300x group velocity slowdown which is the first demonstration of dramatic slow light with 2D semiconductor excitons. We have added additional text to the manuscript to emphasize the novelty the reported slow-light effect in a plasmonic structure. We hope that our improved manuscript is found to be suitable for publication.

Comment 2: One of the principle findings in this work is this 1300-fold slowing of the SPP. This

isn't necessarily that surprising considering that strong coupling between a localized and propagating mode will result in a hybrid dispersion with a propagation length and group velocity in between. This is well known and well reported. The fact that the slowing is so large is interesting to some degree, but considering the plasmon (very slow dispersion) is extremely fast and the exciton is effectively localized, this is what would be expected. The authors thus need to make a comparison to other systems to illustrate how large is this in contrast to the broader field of polaritonic strong coupling (e.g. vibrational polaritons I would assume would have even larger reductions) and then clearly state why this is beneficial other than generally stating these systems are good for on-chip delay lines and buffers.

Response 2: We appreciate the reviewer's concern of the novelty of our system compared to others. The reviewer is correct that vibrational polaritons can potentially have even larger reductions as it was predicted that acoustic phonon polaritons at a hBN-metal interface can experience reductions of up to 50,000 and phonon polaritons up to 10,000⁵. However, when we did an investigation of actual measurements of phonon group velocities, we only found reductions of up to 500^{6,7} on phonons in the 6-12 μm range. We do agree that we need to make the comparisons clear to readers and have included new text in the manuscript (page 8) directly comparing our results to those of our peers. See our response to Reviewer 3 Comments 3,4 for a more direct comparison of our work to those in the community.

Comment 3: In the caption for Figure 1 has two very different descriptions for panel d and calls to the cross polarized spectra that doesn't seem to be there.

Response 3: We thank the reviewer for catching our mistake. In a previous revision of the manuscript Figure 1d included both polarization cases but was changed to only include the co-polarized case. The Figure 1 caption has been corrected to reflect the current version.

Comment 4: I think the addition of propagating character is the more interesting finding. This of course is also not surprising to see, but much like SPP-Epsilon-near-zero coupling in prior works, this offers propagating character to a mode that is otherwise static, while still retaining its pertinent character. I would advise that this should be amplified in this discussion.

Response 4: We thank the reviewer for pointing out the importance of this fact. We have always taken this characteristic for granted as we have always been focused on studying the nonlinear properties of the system. We have added text to the top of page 3 to amplify that we are coupling the exciton to the propagating SPP.

Comment 5: In Fig. 2c there is an asymmetry in the simulated detuning spectrum of the co-polarized configuration that does not show up in the experiment. Is there some insight into the origin of this? It doesn't seem to be simply an artifact of simulations since it is not present in Fig. 2f.

Response 5: The asymmetry in Fig. 2c is related to the fact that we have a hybrid exciton/surface plasmon (E-SPP) mode. Since we choose our parameters such that the pump is exactly at the

center of the exciton line, and the exciton line is taken to be symmetric, the pump-induced change of the susceptibility, $\Delta\chi(\omega)$ is also symmetric. This leads to the symmetric line in Fig. 2f. However, the quantity that determines the line shape of the E-SPP is not directly the excitonic $\Delta\chi(\omega)$, but the change in the propagation wave vector $\Delta k(\omega)$ of the E-SPP, in which the exciton is coupled to the SPP. Assume we are comparing two frequencies, one slightly below and one slightly above the exciton line center. Then the one slightly above the exciton is closer to the SPP resonance than the one slightly below the exciton. This is the physical origin of the anisotropy. Formally, this can be expressed in terms of the proportionality factor that relates $\Delta k(\omega)$ to $\Delta\chi(\omega)$. In the Supplement of Ref. 6, we called this factor h_{env} , it is given there as Eq. 31.

The experimental data in the main text is fairly symmetric; however, this is not always the case. In CPO measurements on other devices, the SPP probe signals can be more asymmetric. There are even small changes to the symmetry of the signal based on the alignment of the beams into the gratings. It is very likely that the nuances of the device fabrication, material defects (our sample is clearly slightly doped), and measurement techniques are to blame for the disagreement between the lineshapes of the experiment and the theory. We have added discussion to the bottom of page 8 summarizing these points.

Comment 6: Finally, I understand that the authors are using the slow down factor to describe the reduction in the V_g in the hybrid SPP-E mode with respect to the uncoupled SPP, while the increase in the propagation length of the hybrid mode wrt the bare exciton. However, I don't believe as written this is going to be clear to the reader. I would advocate for the authors to either present a figure with the propagation lengths and group velocities for all three (bare SPP, bare exciton, hybrid mode) within the frequency range of interest and/or describe in the text how the hybridization provides some averaging of the two bare modes. As written it comes across as the group velocity is reduced and the propagation length increases, which are two contradictory statements without the stated context.

Response 6: First to clarify, our definition of the propagation length is the interaction length of the excitation field and the exciton. In the case of an optical measurement, laser beam normal to the layer, this length would be the atomic height (or thickness) of the crystal (~ 0.7 nm for WSe_2), whereas in the SPP case the length is effectively equivalent to the dimension of the crystal in the dimension of propagation (~ 3 μm in this particular device). The propagation length we discuss should not be taken as related to the group velocity in a $L=v_g t$ sense, but as the length in which the interaction can occur.

Regarding the question bare SPP vs bare exciton vs hybrid, we would first like to note that the bare SPP will not show any effect of coherent population oscillations and thus no reduction in the group velocity since this effect arises from the interaction with the exciton. We are far detuned from the actual SPP resonance and therefore have negligible nonlinearity due to the SPP. The background SPP group velocity will not be changed significantly, it is approximately $0.9c$ (where c is the speed of light in vacuum), which can be inferred from the new figure that we include in the revised version of the Supplement [Supplementary Figure 5b], and also in this response letter (see our answer to Comment 2 of Reviewer 1).

In order to get a spectrally narrow resonance in the nonlinear response, we need to be within the exciton resonance, and therefore either the bare exciton or the hybrid mode will, at least in principle, provide a reduction in v_g . It is, however, important to keep in mind that the exciton alone will not give us propagation along the direction of the layer. In an optical measurement (beam normal to the layers) a TMD monolayer can only affect the velocity with which the light pulse traverses the layer in the direction normal to the layer. In other words, the propagation distance normal to the layer, affected by the bare exciton, is extremely small, only of the order of a nanometer (the layer thickness).

The E-SPP hybrid mode, in contrast, offers propagation distances much longer than that (order of micrometer), because it is localized in the direction transverse to the TMD monolayer and propagates along the layer. This means that the advantage of the hybridization is not getting a sharper resonance, as can be seen by comparing Fig. 2b with 2e, but gaining the possibility of long-distance (>micrometer) propagation lengths. See also our response to Comment 1 of Review 3. We have added discussion to page 8 of the text to clarify these points.

Reviewer #3 (Remarks to the Author):

Comment 1: This paper reports the slow light effect in a 2D semiconductor-plasmonic hybrid system. It's an interesting work. It claims the first demonstration of slow light in 2D semiconductors. The paper can be published given the authors address the following properly.

Response 1: We appreciate the Reviewer's opinion on the viability of work in Nature Communications and hope that our revised manuscript will address their comments.

Comment 2: Why are the excitons in 2D semiconductors special for the slow light purpose? Does it require large exciton binding energy, or sharp resonances with long lifetime? From my understanding, it seems that only a sharp resonance is necessary to have the slow light and there are many possible candidates for such resonances, not limited to excitons.

Response 2: Our answer to this question will, in part, repeat our answer to Comment 5 of Reviewer 2. The Reviewer is of course correct in that a sharp resonance is necessary for slow light. We stress that the width of the resonance in our case is completely unrelated to the 10 meV width of the exciton resonance. It is also unrelated to the exciton binding energy. In coherent population oscillations, the width of the resonance is given by the lifetime of the long-lived excitons, which, in a perfect crystal, are excitons outside the radiative cone or, if the crystal is not perfect and has some impurities or imperfections, localized excitons. But having a narrow spectral line is only a necessary condition for harvesting slow light effects, it is not sufficient.

The other aspect of our work is the geometry of the propagating E-SPP. To understand the geometry aspect, we first note that the benefits of slow light requires a sufficiently long propagation distance. In an optical measurements, if light traverses a TMD monolayer or a GaAs quantum well in the direction normal to the layer or QW, the propagation distance is extremely small (0.7 nm in TMD, ~10 nm in the QW). Unless we are dealing with propagation in

a bulk system, we need optical modes that are propagating along the TMD layer (or along the QW), rather than perpendicular to it. This then requires the mode to be localized in the direction perpendicular to the layer. Herein lies the benefit of the SPP mode, since this mode is localized at the metal surface. If we want the benefits of the narrow line provided by the exciton resonance together with the benefits of the localized mode, we need to bring the TMD monolayer or QW close to the metal surface. The question then arises, is there a difference between bringing the TMD monolayer or a QW close to the surface. The main difference is in how close we can bring the TMD/QW to the surface. Since the thickness of the QW is about 10 nm, it cannot be brought closer than 10 nm to the surface. One might think that there is no large difference between a 1 nm or 10 nm distance of the exciton layer to the metal surface. But we have shown in Ref. 6, that the mode hybridization decreases exponentially with increasing distance, see Eq. 18 of the Supplement to Ref. 6. This is one aspect why, in practice, the TMD monolayer is better than a GaAs QW.

So, while a sharp exciton resonance is a necessary condition, there are other considerations, like the proximity of the electromagnetic mode to the surface, that are important to make the slow-light effect work in practice. We have added further discussion to the main text clarifying the importance of the E-SPP geometry to the top of page 3.

Comment 3: How about IR active phonons for the slow light application? For instance, BN phonons.

Response 3: We note that IR active phonons in BN are also interesting candidates to achieve strong optical effects. When it comes to BN IR phonons in particular, we looked to the works of Ambrosio et al.⁶ and Yoxall et al.⁷ where slow down factors of ~500 are found for ~120 nm thick pieces of hBN. This disadvantages of using a system like this compared to ours is that their propagation length is ~900 nm (compared multiple micron in ours) and that they require N-SOM excitations (where we use simple laser illumination). Also since our slow light effect is optically gated (controlled by the presence of the pump laser), it can potentially be used for dynamic modulation of the group velocity. We have added references to these works to pages 8,9.

Comment 4: Can the authors comment on the excitons in carbon nanotubes(1d system)for the slow light purpose? Can they achieve similar result as the authors demonstrated with 2D materials?

Response 4: After a literature search, we have been unable to find any groups who have measured carbon nanotubes for slow light purposed. There is a theory paper by Li et al.⁸ that uses a very similar pump-probe slow light setup as us. In this article they find a slowdown of ~200 almost an entire order of magnitude below ours. There is an additional theory paper by Solookinejad⁹ that uses a semiconducting carbon nanotube as a quantum dot. They show reductions in the group velocity of 10^4 or 10^5 whether they include or exclude their spin-orbit coupling parameter. These articles ignore the experimental challenges of their systems and the measured values are likely to be even less than they predict.

We infer that generally the Reviewer's intent for both this comment and the previous comment is how does our work compare to that of others. We have previously commented on the Reviewer's recommendations of BN and carbon nanotubes, but we think it is also valid to compare our results to the slow light phenomena in general. In our manuscript we base our measurements off of what Ku et al.¹⁰ did with GaAs/AlGaAs quantum wells where they measure a slowdown of up to 30,000 in a structure of 15 quantum wells. In a review article on slow light in photonic crystals by Baba¹¹ the general maximum values of the group index (what we call the slowdown) found is on the order of 100s. The largest reduction in the group velocity seen so far is on the order of 10^7 which occurred in an atomic gas at μK temperatures¹², while obviously an impressive achievement the difficulties in implementing such an approach for an actual device are tremendous if not impossible.

When comparing our work with what has been done across the spectrum, we would say that we compare quite favorably. However, this fact has not been well stated in the manuscript based on the comments of the reviewers, as such we have added text on pages 8,9 that compares our results to the broader community.

Comment 5: For TMDCs, monolayer is a direct band gap material, but two or multilayer become indirect. However, they still have excitons at K-point. Are those excitons suitable for the experiment in the paper?

Response 5: This is an interesting question, and we think it deserves more research in the future. At this point, we can say that the width of the resonance in the case of excitons at the direct gap is related to the recombination rate of the long-lived populations, which, again, can be excitons outside the radiative cone, or, if the crystal is not perfect, localized excitons. The indirect-gap exciton, being not at the lowest energy, can decay before recombining, and we therefore expect that the rate at which indirect excitons relax to lower-energy excitons, possibly forming direct exciton, will determine the width of a possible population oscillation resonance. If this relaxation is faster than the recombination of the direct-gap exciton, the slow-light effect due to population oscillations will be inferior compared to that of the direct-gap excitons. Investigating these processes would certainly be interesting, but is unfortunately beyond the scope of our present study.

References:

1. Chu, S. & Wong, S. Linear Pulse Propagation in an Absorbing Medium. *Phys Rev Lett* **48**, 738–741 (1982).
2. Schaibley, J. R. *et al.* Population Pulsation Resonances of Excitons in Monolayer MoSe₂ with Sub-1 μeV Linewidths. *Phys Rev Lett* **114**, 137402 (2015).
3. Liu, W., Wang, G., Wen, K., Hu, X. & Qin, Y. Efficient unidirectional SPP launcher: coupling the SPP to a smooth surface for propagation. *Opt Lett* **47**, 621 (2022).

4. Fang, Z. *et al.* Plasmonic Coupling of Bow Tie Antennas with Ag Nanowire. *Nano Lett* **11**, 1676–1680 (2011).
5. Yuan, Z. *et al.* Extremely Confined Acoustic Phonon Polaritons in Monolayer-hBN/Metal Heterostructures for Strong Light–Matter Interactions. *ACS Photonics* **7**, 2610–2617 (2020).
6. Ambrosio, A. *et al.* Selective excitation and imaging of ultraslow phonon polaritons in thin hexagonal boron nitride crystals. *Light Sci Appl* **7**, 27 (2018).
7. Yoxall, E. *et al.* Direct observation of ultraslow hyperbolic polariton propagation with negative phase velocity. *Nat Photonics* **9**, 674–678 (2015).
8. Li, J.-J. & Zhu, K.-D. Tunable slow and fast light device based on a carbon nanotube resonator. *Opt Express* **20**, 5840 (2012).
9. Solookinejad, G. Slow light propagation in carbon nanotube quantum dot with spin-orbit interaction. *Physica B Condens Matter* **547**, 97–100 (2018).
10. Ku, P.-C. *et al.* Slow light in semiconductor quantum wells. *Opt Lett* **29**, 2291 (2004).
11. Baba, T. Slow light in photonic crystals. *Nat Photonics* **2**, 465–473 (2008).
12. Hau, L. V., Harris, S. E., Dutton, Z. & Behroozi, C. H. Light speed reduction to 17 metres per second in an ultracold atomic gas. *Nature* **397**, 594–598 (1999).

REVIEWERS' COMMENTS

Reviewer #1 (Remarks to the Author):

The authors have responded in some detail to the questions and comments of all three reviewers. Since also the other two reviewers seem to agree that the paper is suitable for publication , I now recommend it for publication in Nature Communications.

Reviewer #2 (Remarks to the Author):

First, I apologize for my tardiness. I have reviewed the updated manuscript and response letter and thank the authors for their careful attention to the multiple concerns and comments we raised. In its current form I believe the paper is now suitable for publication.

Reviewer #3 (Remarks to the Author):

I read the response and the revised paper. I am satisfied with the revision. It can be accepted now.